# Docking and Molecular Dynamic of Microalgae Compounds as Potential Inhibitors of Beta-Lactamase

**DOI:** 10.3390/ijms23031630

**Published:** 2022-01-31

**Authors:** Roberto Pestana-Nobles, Yani Aranguren-Díaz, Elwi Machado-Sierra, Juvenal Yosa, Nataly J. Galan-Freyle, Laura X. Sepulveda-Montaño, Daniel G. Kuroda, Leonardo C. Pacheco-Londoño

**Affiliations:** 1Life Science Research Center, Universidad Simón Bolívar, Barranquilla 080002, Colombia; roberto.pestana@unisimon.edu.co (R.P.-N.); yani.aranguren@unisimonbolivar.edu.co (Y.A.-D.); elwi.machado@unisimonbolivar.edu.co (E.M.-S.); juvenal.yosa@unisimonbolivar.edu.co (J.Y.); nataly.galan@unisimonbolivar.edu.co (N.J.G.-F.); 2Department of Chemistry, Louisiana State University, Baton Rouge, LA 70803, USA; lsepcl1@lsu.edu (L.X.S.-M.); dkuroda@lsu.edu (D.G.K.)

**Keywords:** β-lactamase, metabolites, microalgae, docking, molecular dynamic, inhibitors

## Abstract

Bacterial resistance is responsible for a wide variety of health problems, both in children and adults. The persistence of symptoms and infections are mainly treated with β-lactam antibiotics. The increasing resistance to those antibiotics by bacterial pathogens generated the emergence of extended-spectrum β-lactamases (ESBLs), an actual public health problem. This is due to rapid mutations of bacteria when exposed to antibiotics. In this case, β-lactamases are enzymes used by bacteria to hydrolyze the beta-lactam rings present in the antibiotics. Therefore, it was necessary to explore novel molecules as potential β-lactamases inhibitors to find antibacterial compounds against infection caused by ESBLs. A computational methodology based on molecular docking and molecular dynamic simulations was used to find new microalgae metabolites inhibitors of β-lactamase. Six 3D β-lactamase proteins were selected, and the molecular docking revealed that the metabolites belonging to the same structural families, such as phenylacridine (4-Ph), quercetin (Qn), and cryptophycin (Cryp), exhibit a better binding score and binding energy than commercial clinical medicine β-lactamase inhibitors, such as clavulanic acid, sulbactam, and tazobactam. These results indicate that 4-Ph, Qn, and Cryp molecules, homologous from microalgae metabolites, could be used, likely as novel β-lactamase inhibitors or as structural templates for new in-silico pharmaceutical designs, with the possibility of combatting β-lactam resistance

## 1. Introduction

Since the penicillin G discovery in 1929 [1], and its subsequent clinical use in 1940, antibiotics have become the “mainstay” in fighting bacterial infections [2]. The success of these molecules prompted the search and development of additional derivatives. This search resulted in β-lactam antibiotics (penicillin, monobactam, carbapenem, and narrow and extended-spectrum cephalosporins) [2]. Today, these represent between 60 and 65% of the total antibiotics market, generating USD 15,000 million annually [3,4].

From a structural point of view, β-lactam antibiotics are characterized by presenting a highly reactive β-lactam ring (conformed by three carbon atoms and one nitrogen). Physiologically, β-lactam antibiotics inhibit bacterial cell wall biosynthesis through covalent binding and acylation of the active site of penicillin-binding proteins (PBP), irreversibly inactivating them; generating a chain reaction that progressively weakens the cell wall, activates autolysins and hydrolases, and inhibits the control of intracellular osmotic pressure, which leads to cell lysis [5].

As with other antibiotics, the extensive use of β-lactams has led to the emergence and spread of resistant strains. In fact, before the discovery of penicillin, Alexander Fleming had already isolated *E. coli, Salmonella enterica,* and *Haemophilus influenzae* resistant to penicillin. [1,5]. Therefore, antibiotic resistance is a natural phenomenon among bacteria, studied for decades, where study of the molecular mechanisms has been fundamental for a better understanding of resistance strategies. In general, bacteria evade β-lactams through the generation of alternative PBPs with reduced affinity [6,7], reduction of cell permeability through mutations in porins [8], overexpression of flow pumps responsible for expelling the antibiotic to the periplasmic space [9], and the production of enzymes capable of breaking the β-lactam ring (β-lactamases) [10], the last one being the most efficient strategy [11]. In the β-lactamases group, there are extended-spectrum β-lactamases (ESBL) and AmpC β-lactamases, which are enzymes encoded by plasmids or chromosomes, and these genes are horizontally transferred [10]. These adaptive processes have generated an “arms race” between medical chemistry and bacterial evolution, which has seen the introduction of new β-lactams and the immediate emergence of new β-lactamases, both by mutation of previously known families, and by dissemination of genes encoding new enzymes [6], becoming a real threat to public health [12].

The primary strategy used to restore the efficacy of β-lactam antibiotics is the combined use of β-lactamase inhibitors (molecules that can bind to the enzyme’s active site) to prevent the antibiotic from being hydrolyzed by the enzyme [13]. Researches in β-lactamase inhibitory molecules began more than 50 years ago, given the emergence and rapid proliferation of microorganisms of clinical importance with penicillin-resistance [14,15,16], such as *Neisseria gonorrhoeae* [17] and *Haemophilus influenzae* [18]. The first molecule from the natural β-lactamase inhibitors search was clavulanic acid (a broad spectrum suicide inhibitor against staphylococcal and enteric penicillinases) [19]. Clavulanic acid acts synergistically with penicillin and cephalosporin against enteric bacteria producing β-lactamase [6]. After the discovery of clavulanic acid, the next generation of β-lactamase inhibitors were synthesized from penicillanic acid sulfones, molecules, such as sulbactam and tazobactam, which were commercialized successfully. However, with the passage of time, their efficacy significantly decreased by the appearance of TEM-1 β-lactamases [20].

In recent years, new groups of inhibitors have appeared, and some have already been approved by regulatory agencies and are now available in the clinical setting, thus extending and recovering the antimicrobial activity of some β-lactam antibiotics [21]. In addition, a new class of non-β-lactam inhibitors has emerged based on a diazabicyclooctane (BOD) [22]—with a broader spectrum than clavulanic acid and sulfones—that inhibit class A penicillinases, ESBLs, class C serine carbapenemases and cephalosporinases, and some class D oxacillinases [23,24]. However, like their predecessors, bacterial resistance against these new molecules has been observed since 2020 [25,26]. Currently, synthesized molecules from boronic acid [27] have been used as the last line of battle against bacterial infections, especially for *Klebsiella pneumoniae* and other carbapenemase-producing enterobacteria, considerably increasing the effect of carbapenems [28]. The search and discovery of new β-lactamase inhibitors is a promising approach to combat the prevalent resistance to β-lactams. However, this approach is challenged by the variable affinity of inhibitors, the amount and variety of β-lactamases produced by resistant bacteria [13], and differences in vitro/in vivo conditions used to determine antibiotic susceptibility [29]. Future inhibitors must be potent and capable of simultaneously inhibiting different classes of β-lactamases [12]. Nowadays, there is no inhibitor with these characteristics; therefore, finding an inhibitor has become indispensable [30], requiring complex structural and biochemical development.

Natural products (NPs) are safe resources for human use and valuable sources of substitutes for medicines. NPs are generally secondary metabolites derived from microorganisms, plants, and animals [31,32]. These molecules are dynamic, and their properties and biosynthesis levels depend on genetic and environmental changes. Microalgae produce an extraordinary variety of secondary metabolites, often unique, and differ from those identified in terrestrial organisms, because they have special metabolic patterns closely linked to the unique characteristics of their environments, consisting, for example, of continuous variations of light, pressure, nutrients, salinity, and temperature [33].

NPs have evolved over millions of years, and they have acquired a unique chemical diversity, resulting in a diversity of their biological activities and drug-like properties. Therefore, even before the rise of modern pharmacology, NPs have been used for centuries as components of traditional medicine. The interest in NPs and their application triggered an almost exponential growth in the number of NP databases, industrial catalogs, bibliographic material, and chemical structures from various organisms and geographic locations [34]. In this sense, an in vitro analysis has been a daunting and inefficient task when looking for a molecule with a specific function (for example, a broad-spectrum inhibitor of β-lactamases). Therefore, computational biology and molecular modeling methods are necessary to accelerate research, save time and money, and obtain better results; over the years, this approach has proven to be a powerful tool in the search for new inhibitors [12,30,35,36,37,38,39,40].

This study was designed to investigate NPs as metabolites present in microalgae to identify new and more efficient biomolecules with potential inhibitory action against β-lactamase. Computational methods, including molecular docking and dynamic simulations of collections of metabolites from microalgae, were explored. Moreover, these were compared with commercial inhibitors.

## 2. Results and Discussion

### 2.1. Molecular Docking

A total of 3652 structures from microalgae metabolites and homologous compounds (M) previously selected were docked against six β-lactamase proteins [41]. A comparison of the binding energy (BE) of the best docked compounds with the three β-lactamase inhibitors [2] introduced by clinical medicine (β-LICM) is represented in Figure 1c. These three β-LICM; clavulanic acid (CA) [42], sulbactam (SB) [43], and tazobactam (TZ) [41] were selected: CA is the first β-lactamase inhibitor introduced into clinical medicine, SB and TZ are penicillinase sulfones that were used later in clinical medicine. These inhibitors are structurally like penicillin, as shown in Figure 1a.

Six structures for M with the lowest BE (M-LWBE) for each β-lactamase are shown in Figure 1b; for the docking with 1NYM the M-LWBE was azaleatin 3-rutinoside (M1), which is a glycoside molecule and a member of flavonoids; for 1C3B, the M-LWBE was quercetin 3-(6’-O-caffeoyl)-beta-d-glucopyranoside (M2), which is a quercetin O-glucoside; for 3V50, the M-LWBE was 6-phenyl-7H-benzo[a]phenalene (M3), which is an aromatic hydrocarbon; and for the docking with 5N5I, 7KHQ, and 4EXS, the M-LWBE were (3S,10R,13E,16S)-10-[(3-Chloro-4-methoxyphenyl)methyl]-16-[(1S)-1-[(2R,3R)-3-[4-(chloromethyl)phenyl]oxiran-2-yl]ethyl]-6,6-dimethyl-3-(2 methylpropyl)-1,4-dioxa-8,11-diazacyclohexadec-13-ene-2,5,9,12-tetrone (M4), cryptophycin A (M5) and 3S,6R,10R,13E,16S)-10-[(3-chloro-4-methoxyphenyl)methyl]-6-methyl-16-[(1S)-1-[(2S,3S)-3-(4-methylphenyl)oxiran-2-yl]ethyl]-3-(2-methylpropyl)-1,4-dioxa-8,11-diazacyclohexadec-13-ene-2,5,9,12-tetrone (M6), respectively. Where M4, M5, and M6 are molecules that belong to the cryptophycin family. 

The comparison between the BE from M-LWBE and β-LICM shows that the BE for M-LWBE was lower than β-LICM, as observed in Figure 1c. This indicates that, from a theoretical point of view, it is possible that these six microalgae metabolites and homologous compounds (M1–M6) could likely be better inhibitors than the three traditional β-LICMs. Additional analysis from the ten (10) structures with the lowest Bes for each β-lactamase protein docking revealed that these molecules have structural similarity among them, i.e., the molecules with the best results for each molecular docking came from the same structural template or homologs of known structures (see Figure 1d). These structures were classified according to the repeated structural template in five homolog families, such as phenylacridine (4-Ph) [44], quercetin (Qn) [45], cryptophycin (Cryp) [31], noscomin, and sytoscalarol, with the Cryp structure being the most persistent [46,47,48] through the different M-LWBE.

These present results are in accordance with the experimental study conducted in 2021 by Alshuniaber MA et al., who reported that 4-Ph is a secondary metabolite found in spirulina with potential antibacterial activity against foodborne drug-resistant bacteria due to polyphenol structural behavior. In the case of Qn, Al-Saif SS, et al. [45], reported that this bioflavonoid is present in a wide variety of marine algae isolated from the coast of the Red sea, such as *Ulva reticulata*, *Caulerpa occidentalis*, *Cladophora socialis*, *Dictyota ciliolata*, and *Gracilaria dendroides,* with different biological activities, including the biological action as an inhibitor of the pathogenic bacteria growth both Gram-positive and Gram-negative, as well as antioxidative tissue-protective and tumoristatic effects and inhibition of hepatic cholesterol biosynthesis [49]. 

However, according to the results of this study, the marine microalgae metabolite Cryp is the most promising molecule, because it is the most common structural template found with the lowest BE in the whole molecules docked. Additionally, in experimental studies, Cryp was already reported as a natural product isolated from blue–green algae with anti-tumor activity in an in-vitro cell analysis [47]; this macrolide depsipeptide has potent cytotoxicity, which gives it an anticancer activity in murine leukemia and carcinoma cell lines [31]. 

Another aspect that made interesting Cryp homologs as a potential β-lactamase inhibitor was the docking behavior observed with all β-lactamase proteins selected, except with the 3V50 β-lactamase protein. As shown in Figure 1d, low binding energies for Cryp docked with 1NYM, 1C3B, 5N5I, 7KHQ, and 4EXS were found. Therefore, Cryp could possibly represent a possible broad-spectrum drug for β-lactam antibiotic-resistant bacteria.

### 2.2. Molecular Dynamic

The initial position for molecular dynamics was set in the molecular docking result. A total of 24 systems (protein–ligand) were analyzed by molecular dynamics. Figure 2a shows the 3D structure of the six β-lactamase proteins (with their respective PDB IDs) docked with the best BE ligand (represented as green sticks). The RMSD variation is used to determine the structural conformation changes and stability in a protein [50]. The RMSD variation during the molecular dynamic reflects a stable behavior for each system with CA, SB, TZ, protein without ligand (P), and M ligands (see Figure 2b). Additionally, there was little difference between the RMSD variation of the proteins without the ligand and the proteins with the ligand. It is indicative that the structure, in general, remains similar, whereby the protein’s function should not be affected.

### 2.3. MMPBSA 

The binding free energy (BFE) represents the average interaction energy between the receptor and the ligand; this was used to reevaluate the initial results obtained by the docking; the advantage of considering BFE is that this method not only consider the protein–ligand contribution, but also takes into count the surroundings (water, ions, etc.). The BFE reported is the sum of contributions of different energies (van der Waals, electrostatic energy, electrostatic contribution to the solvation free energy, and nonpolar contribution to the solvation free energy), where the sum is equal to the final, binding free energy calculated [51].

Figure 3a shows the energy of each system, and are named residues with an energy contribution below −0.4 Kcal/mol. It can be observed how the BFE of the ligands is better in almost all cases compared with β-LICM, and the best β-LICM evaluated is TZ in all cases. The contribution for each residue in the BFE was evaluated through the energy decomposition option available in the MMPBSA Amber script. Figure 3b shows the contributions of binding free energy for each residue in their respective protein. The most favorable interactions are characterized by the lower BFE values (pointed out in Figure 3b). For 1C3B, the most favorable interactions are given by LEU 116 and ALA 217, for 1NYM the GLU 146 residue, for 3V50 the ILE 196 residue, for 5N5I the PHE 31 residue, for 7KQH the LEU 173, and 4EXS the PHE 37 residue.

In Figure 4, one can see the molecular interactions, protein–ligands, for six β-lactamase proteins with the best BE compounds (green sticks) at 5 Å of the ligands. The blue and red residues illustrate the most favorable and unfavorable interactions, respectively. In the case of the 1C3B-M2 complex (Figure 4a), the favorable hydrophobic interactions with LEU 116 and TYR 218, as a strong unfavorable interaction with LYS 64, which is a positive electrically-charged residue, were observed. In the 1NYM-M1 complex (Figure 4b), there were strong hydrophobic favorable interactions with residues (TYR 80 and MET 245). Moreover, unfavorable interactions with positive residue ARG 218 were observed in the 1NYM-M1 complex.

Additionally, favorable hydrophobic interactions with ILE 196 and ALA 253 were observed in the 3V50-M3 complex (see Figure 4c); the order of favorable interaction strength are represented with blue coloration intensity. Moreover, in this complex, an unfavorable interaction with a positive-charged residue, ARG 218, was observed. 

In the 4EXS-M6 (see Figure 4d) complex, only favorable interactions with hydrophobic residues, such as MET 34, PHE 37, and VAL 40, were observed. In Figure 4e, the 5N5I-M4 complex showed favorable hydrophobic interactions with PHE 31 and TYR 36, and an unfavorable interaction with a negative residue, GLU 171, was observed. Finally, in the 7KHQ-M5 complex (see Figure 4f), favorable hydrophobic interactions with ILE 79, TRP 82, LEU 135, and LEU 224 were observed. Further, an unfavorable interaction with a basic residue, ARG 227, was observed.

As the docking results, the Cryp homologous clearly shows a better performance, with a difference of more than 10 Kcal/mol (see Figure 3a) compared with the best β-LICM in 5N5I, 7KHQ, and 4EXS proteins, giving support to the idea of using this molecule as a possible template for in silico design or as a starting point for other methodologies as hit-to-lead [52]. 

## 3. Materials and Methods

### 3.1. Protein Selection

Six β-lactamase proteins were selected, based on the Bush–Jacoby group [55,56]; all protein structures were downloaded from the Protein Data Bank (PDB) [41]. Table 1 presents information for the protein selected (Bush–Jacoby classification, name, bacteria where this protein is expressed, and PDB id). For proteins 1NYM, 5N5I, 4EXS, and 7KHQ, the ions available in their structures were deleted for the docking studies and kept for the molecular dynamics studies. 

### 3.2. Ligands Selection

A total of 15 metabolites of microalgae compounds with bioactivity, plus their homologous, for a total of 3652 structured, were selected for the docking studies. Moreover, 3D structures of all the ligands were downloaded from the PubChem database [63].

### 3.3. Molecular Docking

Initially, for each ligand, the structure was optimized using XTB software [64] at an extreme level using GFN2-xTB [65].

For the proteins in the docking studies, all ions and waters were deleted as well as another possible solvent molecules in the PDB structure; this was achieved through chimera [53].

Docking was carried out to evaluate the interaction between the compounds with the target proteins by measuring the binding energy of the complex. AutoDock Vina 1.2.0 was used for this process through the Vina forcefield [66,67].

The ligands were placed at the positions reported by the inhibitors in their respective PDB files; an “exhaustiveness” of 32 was used. The box was placed at the center of the ligands, the size of the box was 30 Å in each direction, and the default value (0.375 Å) for the spacing was used. The best docking result for each protein was used as a starting point for the molecular dynamics simulation.

The PDBQT for the protein and the ligand for AutoDock Vina was obtained using scripts from the ADRF suite [68] and Meeko library, available at (https://github.com/forlilab/Meeko; accessed on 1 December 2021).

### 3.4. System Preparation for Molecular Dynamics Simulation

Each protein, water, and inhibitor molecule was removed from the original PDB file. The Mulliken charge [66] was calculated for the ligands through antechamber in AmberTools 18; the topology and charge file were obtained through antechamber and parmchk2 [69]. 

For each system (protein–ligand), the ff14SB forcefield for the protein [70] and GAFF forcefield for the ligand were used [71]; the mbondi2 parameters of atomic radii for the generalized Poisson–Boltzmann calculations [72,73,74] were set; the system was solvated in a cube box, putting at least 15 Å from the protein to the edge of the box, water TIP3P was used [75,76], Na+ or Cl− ions were added to neutralize charges. 

### 3.5. Molecular Dynamics (MD)

All molecular dynamics simulations were carried out using Amber 18 [69]; two minimizations were carried out. First, the water molecules were minimized, the maxcyc and ncyc as 50,000 and 1500 were established, respectively, and ntc was 1 (the shake was off); for this, the solute was restrained with a force of 100 kcal/mol-Å^2^. 

Afterward, the entire system was minimized, using maxcyc = 100,000 and ncyc = 1000. After energy minimization, the system was gradually heated until 300 K for 500 ps. In this part, the solute was restrained with a force of 2.0 kcal/mol-Å^2^. Next, a pressure equilibration was carried out for 50 ps. Finally, the systems were equilibrated, 500 ps. For the production, for each system, a total of 10 ns of the simulations were performed. A total of 24 systems were evaluated using MD, each protein with three ligands previously reported as β-lactamase inhibitors (18 systems), and each protein with the best docking results (6 systems). 

### 3.6. MMPBSA

Binding free energy was calculated through MM/PBSA script from the Amber package [51]; for this, the last 2 ns were used as a sample, giving a total of 200 snapshots, considering more than 2 ns can decrease the accuracy of the results as a previous author mentioned [74,77]. Entropy was not considered due to the high computational costs and the possibility of decreasing the accuracy of the MM/PBSA method [78,79,80]. The parameters inp = 1 and radiopt = 0 were stablished; for the energy decomposition, the idecomp option was set to 1.

### 3.7. RMSD Calculation

RMSD calculations were performed using the ccptraj tool [81], using the first frame of production as the reference. 

## 4. Conclusions

Finding new molecules for the inhibition of β-lactamase proteins remains a challenge to be solved in the clinical field, in the face of bacterial resistance.

A theoretical methodology based on molecular docking and molecular dynamics was used to find new β-lactamase inhibitors from microalgal metabolites. Both theorical results (molecular docking and molecular dynamics) revealed that metabolites belonging to the same structural families, such as 4-Ph, Qn, and Cryp, exhibit better binding scores, energy, and affinity with β-lactamase than commercial inhibitors, such as CA, SB, TZ.

These results indicate that 4-Ph, Qn, and Cryp molecules, homologous from microalgae metabolites, could likely be used as novel β-lactamase inhibitors or as structural templates for new in-silico pharmaceutical designs, with the possibility of combatting β-lactam resistance. 

On the other hand, despite Cryp family structures being the most recurrent molecules observed, they are reported as cytotoxic substances in cell lines [82], limiting their use as inhibitors. However, if the necessary concentration for β-lactamase inhibition is less than IC50, they could possibly be used.

## Figures and Tables

**Figure 1 ijms-23-01630-f001:**
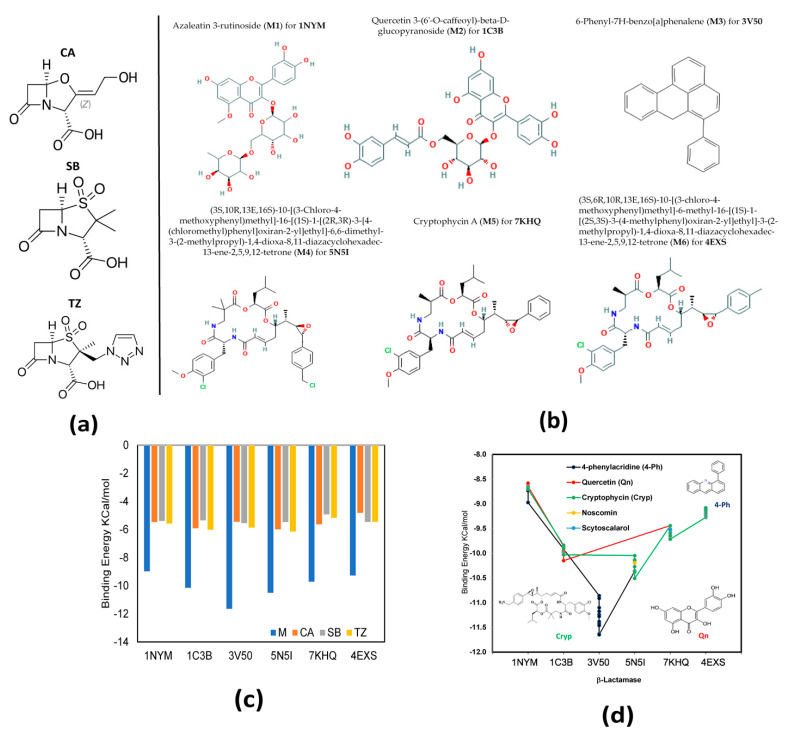
(**a**) Structure of ligands previously reported as β-lactamase inhibitors; (**b**) best docked ligands with lowest interaction energy; (**c**) plot of binding energy comparison between best docked ligand and β-lactamase inhibitors; and (**d**) plot of the best binding energy from the ten (10) docking for each one of the β-lactamase proteins selected and their structural similarities.

**Figure 2 ijms-23-01630-f002:**
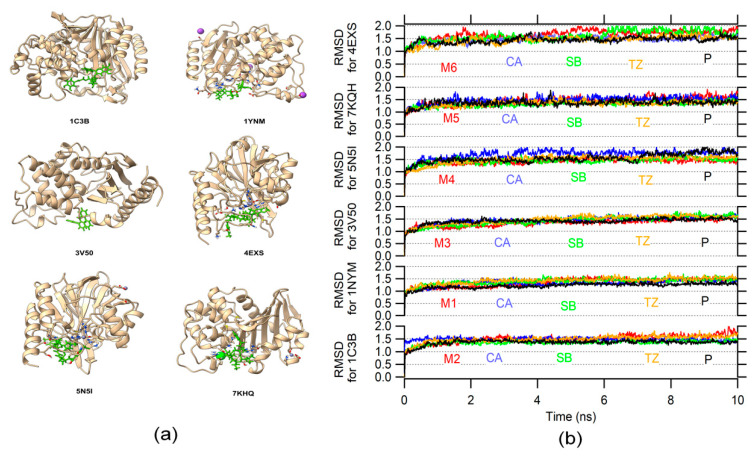
(**a**) Structure of the six β-lactamase proteins (with their respective PDB IDs) docked with the best BE ligand (represented as green sticks), the green, blue, and purple spheres are chloride, zinc, and potassium atoms, respectively. (**b**) The RMSD variation during the molecular dynamic for each β-lactamase with the six ligand structures (M1–M6); the three β-LICM binding poses, and β-lactamase without the ligand (P).

**Figure 3 ijms-23-01630-f003:**
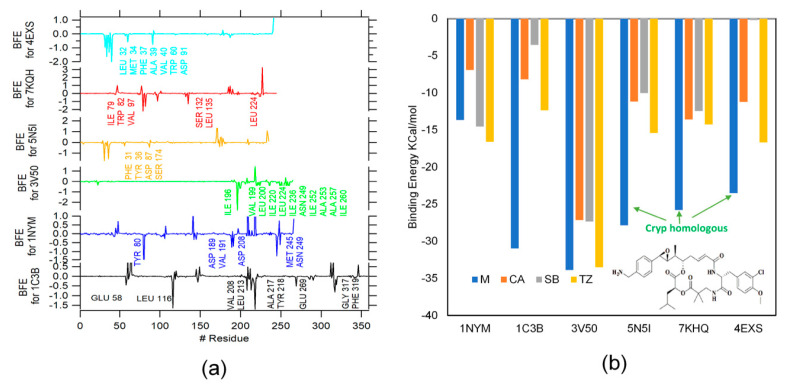
(**a**) A plot of binding free energy for each residue from β-lactamase proteins, residues with energy contributions below −0.4 Kcal/mol are marked. (**b**) The binding energy for the ligand and inhibitors by dynamic molecular.

**Figure 4 ijms-23-01630-f004:**
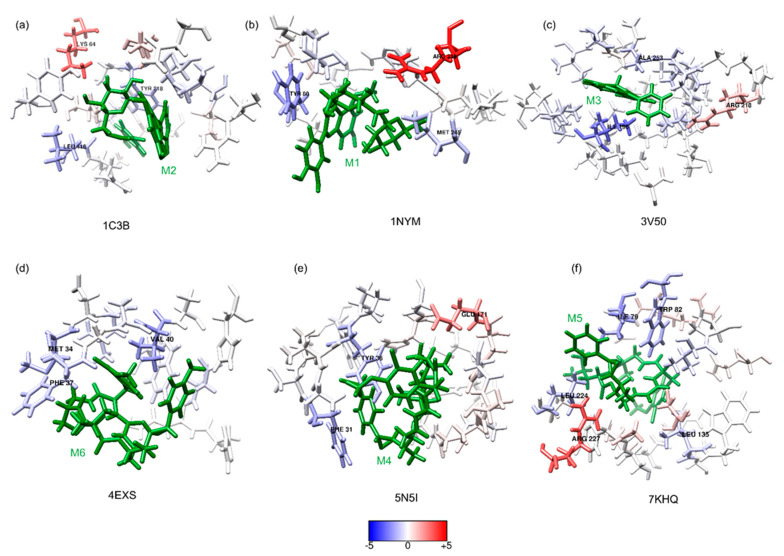
Energy contribution per residue of the six β-lactamase proteins around 5 Å of the ligands (represented as green sticks), the blue and red residues illustrate the favorable and unfavorable interactions, respectively; residues with a higher than 1 and lower than −1 are named. (**a**) Complex 1C3B-M2, (**b**) 1NYM-M1, (**c**) 3V50-M3, (**d**) 4EXS-M6, (**e**) 5N5I-M4, and (**f**) 7KHQ-M5. This figure was obtained using the chimera plugin, CHEWD [53,54].

**Table 1 ijms-23-01630-t001:** Information about the six β-lactamase proteins selected.

Bush–Jacoby Group (2009)	Name	Bacteria	PDB Protein Data Bank
1	AMPC beta-lactamase	*Escherichia coli*	1C3B [57]
2b	M182T mutant of TEM-1	*Escherichia coli*	1NYM [58]
2br	Complex of SHV S130G mutant beta-lactamase complexed to SA2-13	*Klebsiella pneumoniae*	3V50 [59]
3a	VIM-1 metallo-beta-lactamase	*Pseudomonas aeruginosa*	5N5I [60]
2df	OXA-48 K73A	*Klebsiella pneumoniae*	7KHQ [61]
1	NDM-1	*Klebsiella pneumoniae*	4EXS [62]

## Data Availability

Not applicable.

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
