# Peer review of "Docking and Molecular Dynamic of Microalgae Compounds as Potential Inhibitors of Beta-Lactamase"

_ijms, 2022, doi:10.3390/ijms23031630_

Round 1
Reviewer 1 Report
clear 2,564 / 5,000Resultados de traducción
1. The abstract is adequate and contains the relevant elements. 2. The introduction is concise and clear. The problem is mentioned in a way clear and the most relevant and updated antecedents are cited. 3. The materials and methods are presented the most relevant details that allow replicating simulations. The proposed methodology is sufficient and updated to answer the research question. 4. The analysis of outcomes is adequate with the results obtained. Thedescriptions of the interactions and their connection to the stabilities of the
systems is very good at understanding the different affinities. The discussion of the diferent molecules provides an interesting aspect of the flexibility of the ligand and how this influences the interactions that occur they form between the ligand and the protein.
5. The conclusions respond to the objectives of the work and are based
in the results obtained. The scope of the conclusions agrees
with the possibilities of the methods used, but perspectives are proposed
for future jobs.
6. The work is novel in terms of the ligands and protein studied. The
used methods, although they are not new are reliable, which guarantees
that the results obtained are reproducible. On the other hand, the issue studied and the approach given is interesting and brings a new knowledge in the study of this type of molecules.
7. The style and writing are consistent with the quality of a scientific article.
8. Tables and figures are relevant and sufficient. Information shown contributes to the discussion and are presented in the place of the text that
makes your inquiry easy. I recommend reviewing Figure 1, since it is relevant for the presentation of the study is presented with poor quality and in a size that makes it difficult to interpret.
9. References are adequate and sufficient. I suggest reviewing the works
Recent Ahumedo et al. who use tools and analyzes similar to
a protein-ligand interaction problem. For example: Journal of
Molecular Graphics and Modeling, 86, 2019, 113-124. and Mol. Biosyst. 2014,
10, 1162-1171. Quantum mechanics study of the hydroxyethylamines–BACE-1 active site interaction energies C Gueto-Tettay, JC Drosos, R Vivas-Reyes Journal of computer-aided molecular design 25 (6), 583-597 2011CoMFA, LeapFrog and blind docking studies on sulfonanilide derivatives acting as selective aromatase expression regulators C Gueto, J Torres, R Vivas-Reyes European journal of medicinal chemistry 44 (9), 3445-3451
Author Response
Figure 1 was improved
The References recommended by the revisor were revised and add

Reviewer 2 Report
Manuscript entitled „Docking and Molecular Dynamic of Microalgae Compounds as Potential Inhibitors of Beta-Lactamase” describes an interesting model for antimicrobial resistance. I found it interesting but I have major concerns about some experiments. However I think that authors will be able to explain them and revise manuscript.
The paragraph between 230-239 is not clear can be rewrite in more clear format.
Figure 1 is hard to read reformat and rearrange!
The conclusion part need to rephrase. My recommendation is to focus on 3 short conclusion which can have practical use in clinical field!
Please double check the article by a native English reader.
Please recheck the References order.
Moreover editing of manuscript including grammar check is required.
Author Response
The paragraph between 230-239: “Likewise, in the 3V50-M3 complex (see Fig 4(c)), favorable hydrophobic interactions with ILE 196, and ALA 253, which are mentioned in order of strength, as an unfavorable interaction with a positive charged residue ARG 218 were observed. In the 4EXS-M6 complex, only favorable interactions were observed; interactions with hydrophobic residues such as MET 34, PHE 37, and VAL 40 are shown in Figure 4(d). In the 5N5I-M4 complex (see Fig 4(e)), favorable hydrophobic interactions with PHE 31 and TYR 36 unfavorable interaction with a negative residue GLU 171 were observed. Finally, in the 7KHQ-M5 complex (see Fig 4(f)), favorable hydrophobic interactions with ILE 79, TRP 82, LEU 135, and LEU 224, and were observed. Further, unfavorable interactions with diverse types of residues such as a basic residue ARG 227 were observed”.
It was rewrite:
“Additionally, favorable hydrophobic interactions with ILE 196, and ALA 253 were observed in the 3V50-M3 complex (see Figure 4(c)), the order of favorable interactions strength are represented with the blue coloration intensity. Also, in this complex an unfavorable interaction with a positive charged residue ARG 218 was observed.
In the 4EXS-M6 (see Figure 4(d)) complex, only favorable interactions with hydrophobic residues such as MET 34, PHE 37, and VAL 40 were observed. In Figure 4(e), the 5N5I-M4 complex shown favorable hydrophobic interactions with PHE 31 and TYR 36, and unfavorable interaction with a negative residue GLU 171 was observed. Finally, in the 7KHQ-M5 complex (see Figure 4(f)), favorable hydrophobic interactions with ILE 79, TRP 82, LEU 135, and LEU 224 were observed. Further, an unfavorable interaction with a basic residue ARG 227 was observed”.
The Conclusion was rephrased and modified.
“Finding new molecules for the inhibition of β-Lactamase proteins remains a challenge to be solved in the clinical field, in the face of bacterial resistance.
A theoretical methodology based on molecular docking and molecular dynamics was used to find new β-Lactamase inhibitors from microalgal metabolites. Both theorical results (molecular docking and molecular dynamics), revealed that metabolites belonging to the same structural families such as 4-Ph, Qn, and Cryp exhibit better binding scores, energy, and affinity with β-Lactamase than commercial inhibitors as CA, SB, TZ.
These results indicate that 4-Ph, Qn, and Cryp molecules and homologous from microalgae metabolites could be used likely as novel β-Lactamase inhibitors or as structural templates for new in-silico pharmaceutical designs with the possibility to combat the β-lactams resistance.
On the other hand, despite Cryp family structures being the most recurrent molecules observed, they are reported as cytotoxic substances in cell lines [82], limiting their use as inhibitors. However, if the necessary concentrations for β -lactamase inhibition is less than IC50, they could possibly be used.
